# In Vivo Evidence of Melatonin’s Protective Role in Alkylating-Agent-Induced Pulmonary Toxicity: A Systematic Review

**DOI:** 10.3390/antiox14060712

**Published:** 2025-06-11

**Authors:** Emma Sola, Jose A. Morales-García, Francisco López-Muñoz, Eva Ramos, Alejandro Romero

**Affiliations:** 1Department of Pharmacology and Toxicology, Faculty of Veterinary Medicine, Complutense University of Madrid, 28040 Madrid, Spain; esola@ucm.es; 2Department of Cell Biology and Histology, School of Medicine, Complutense University of Madrid, 28040 Madrid, Spain; jmoral06@ucm.es; 3Faculty of Health Sciences-HM Hospitals, University Camilo José Cela (UCJC), Villafranca del Castillo, 28692 Madrid, Spain; flopez@ucjc.edu; 4HM Hospitals Health Research Institute, 28015 Madrid, Spain; 5Neuropsychopharmacology Unit, Hospital 12 de Octubre Research Institute, 28041 Madrid, Spain

**Keywords:** melatonin, lung toxicity, alkylating agents, human equivalent doses, systematic review, oxidative stress, inflammation, histopathology

## Abstract

Alkylating agents, historically employed as chemical warfare agents and currently used as chemotherapeutic drugs, are known to induce significant pulmonary toxicity. Current clinical interventions often fail to fully prevent or reverse these pathological changes, highlighting the urgent need for safe, broad-spectrum therapeutic agents that are effective across diverse exposure scenarios. Melatonin has emerged as a promising protective agent due to its antioxidant, anti-inflammatory, and immunomodulatory properties, along with a well-established safety profile. This systematic review evaluates the potential of melatonin in mitigating vesicant-induced pulmonary damage, synthesizing and critically analyzing preclinical evidence in accordance with the PRISMA guidelines. Three in vivo rodent studies met the inclusion criteria and were analyzed. In all cases, melatonin demonstrated protective effects against alkylating agents such as mechlorethamine (HN2) and cyclophosphamide (CP). These effects were dose-dependent and observed across various animal models, administration protocols, and dosages (ranging from 2.5 to 100 mg/kg), highlighting the importance of context-specific considerations. The human equivalent doses (HEDs) ranged from 12 to 973 mg per day, suggesting that the effective doses may exceed those typically used in clinical trials for other conditions. Melatonin’s pleiotropic mechanisms, including a reduction in oxidative stress, the modulation of inflammatory pathways, and support for tissue repair, reinforce its therapeutic potential in both prophylactic and treatment settings for alkylating agent exposure. Nonetheless, this review underscores the critical need for further randomized clinical trials to establish the optimal dosing strategies, refine treatment protocols, and fully elucidate melatonin’s role in managing alkylating-agent-induced pulmonary toxicity.

## 1. Introduction

Cancer remains one of the leading causes of morbidity and mortality worldwide, driving the ongoing search for more effective therapeutic strategies. The introduction of nitrogen mustards as alkylating agents in the early 1940s marked a significant milestone in oncology, ushering in the era of modern cancer chemotherapy [1,2]. Originally developed for chemical warfare agents due to their potent vesicant properties, alkylating agents have retained clinical relevance owing to their cytotoxic effects on rapidly dividing cells [3,4]. However, their lethality and long-lasting adverse effects soon became evident, as individuals, particularly military personnel, exposed to these agents experienced severe skin blistering, ocular damage, and, most critically, acute and chronic respiratory tract injuries.

In the ensuing decades, research into the mechanisms of toxicity led to the clinical development of nitrogen mustards, a subclass of alkylating agents. Initially recognized for their ability to deplete leukocytes, nitrogen mustards were later repurposed as antitumor agents following early evidence of efficacy in treating malignant lymphoma [2].

Despite their therapeutic value, the clinical use of nitrogen mustards is significantly limited by their off-target toxicity, particularly within the lungs. Due to their large surface area, dense vascularization, and thin alveolar–capillary barrier, the lungs are especially susceptible to damage. Alkylating agents can cause both acute and chronic pulmonary injury through mechanisms such as DNA alkylation, oxidative and nitrosative stress, inflammation, mitochondrial dysfunction, and programmed cell death [5,6,7]. Pulmonary complications, including edema, fibrosis, and severe inflammation, are commonly reported, and the current treatment options often fail to prevent or reverse these pathological changes [7,8]. Nevertheless, nitrogen mustards such as mechlorethamine, chlorambucil, melphalan, and cyclophosphamide remain foundational in the treatment of various hematological malignancies and solid tumors [9,10,11,12]. They continue to play a critical role in oncology protocols, both as standalone agents and as precursors to newer drugs, underscoring their enduring clinical importance and the pressing need for improved safety profiles [2].

A major concern is the lack of effective prophylactic or disease-modifying interventions to prevent or reverse the fibrotic remodeling that characterizes chronic lung injury. Given the persistent risk of exposure in military, industrial, and civilian settings, there is an urgent need for novel therapeutic agents that are both safe and effective across a range of exposure scenarios.

Melatonin, a pleiotropic indoleamine best known for its role in regulating circadian rhythms, has gained increasing attention as a potential countermeasure against alkylating-agent-induced toxicity [13,14,15]. Its potent antioxidant, anti-inflammatory, and immunomodulatory properties, along with its ability to modulate mitochondrial function and epigenetic pathways, suggest broad therapeutic potential in mitigating vesicant-induced pulmonary damage [13]. Preclinical studies have demonstrated that melatonin administration reduces oxidative stress markers, restores endogenous antioxidant enzyme activity, and ameliorates histopathological lung injury in animal models exposed to agents such as mechlorethamine [16] and cyclophosphamide [17]. These protective effects are largely attributed to melatonin’s capacity to scavenge a wide range of reactive oxygen and nitrogen species (RONS), including hydroxyl radicals, peroxynitrite, singlet oxygen, and nitric oxide [18]. Additionally, melatonin enhances the activity of key antioxidant enzymes such as superoxide dismutase (SOD), glutathione peroxidase, and catalase, thereby strengthening cellular defense mechanisms [19].

Beyond its antioxidant capacity, melatonin also exhibits strong anti-inflammatory effects. It suppresses the expression of pro-inflammatory cytokines (e.g., TNF-α, IL-1β, and IL-6), inhibits NF-κB signaling, and downregulates the activation of inflammasomes such as NLRP3, which are implicated in the pathogenesis of chemically induced pulmonary inflammation and fibrosis [20].

Importantly, melatonin has an excellent safety profile, even at high doses, making it an attractive candidate for further exploration as a prophylactic or adjunctive therapy, particularly in high-risk environments or when conventional treatment options are limited [21]. Despite promising preclinical findings, several translational challenges remain, including the optimization of dosing regimens and the validation of its efficacy in clinical settings.

This systematic review aimed to critically evaluate the current evidence regarding melatonin’s protective role against alkylating-agent-induced pulmonary toxicity, identify key knowledge gaps, and offer recommendations for future research and clinical applications.

## 2. Materials and Methods

Systematic reviews and meta-analyses serve as precise and reliable tools in evidence-based practice. They establish a gold standard for gathering and synthesizing evidence across diverse subjects, allowing for the resolution of contradictory findings and supporting the continuous updating of knowledge. These approaches significantly enhance professional decision-making by reducing bias and minimizing misinterpretations.

In this context, we conducted selection and a bibliographic analysis following the systematic review methodology outlined in the PRISMA statement [22]. All authors contributed to the design and validation of the review protocol. Specifically, following the PRISMA guidelines, we compiled an initial pool of studies, assessed the quality and relevance of each study to the topic of interest, extracted key data, and performed a critical synthesis. A 27-item checklist, covering the introduction, materials and methods, results, and discussion sections, was developed based on the PRISMA2020 statement [23].

This methodology was chosen to identify all the in vivo studies investigating the co-treatment of melatonin and alkylating agents published up to 10 March 2025. The adoption of the PRISMA protocol ensured transparency and reproducibility in both the selection criteria and the analytical procedures. The detailed, step-by-step methodology, aligned with the PRISMA framework, is presented below. This systematic review was registered in Open Science Framework (OSF) (registration DOI: 10.17605/OSF.IO/MWKU6) [24].

### 2.1. Search Strategy

A systematic literature search was conducted across major biomedical databases: PubMed/Medline (https://pubmed.ncbi.nlm.nih.gov, accessed on 10 March 2025), Scopus (https://www.scopus.com, accessed on 10 March 2025), Web of Science (https://www.webofscience.com/wos/woscc/basic-search, accessed on 10 March 2025), and the Cochrane Library (https://www.cochranelibrary.com, accessed on 10 March 2025). Relevant studies were identified using predefined key terms and Boolean operators, combined in the following search algorithm: (((melatonin[Title/Abstract]) AND (Cyclophosphamide OR Chlorambucil OR uramustine OR mechlorethamine OR melphalan OR bendamustine)) NOT (Review[Publication Type])) AND (English[Language]).

When applicable, medical subject heading (MESH) terms were employed to enhance the identification of studies specifically addressing the review topic. To ensure comprehensive coverage, the electronic search was supplemented with manual hand-searching of reference lists.

### 2.2. Inclusion and Exclusion Criteria

Only original research articles published in English were included; review articles were excluded. As of 10 March 2025, the eligible studies met the following criteria:-Focused on in vivo experimental models;-Involved the co-administration of melatonin and an alkylating agent;-Specifically evaluated pulmonary toxicity or lung-related outcomes.

Studies not meeting these criteria were excluded from the final analysis.

### 2.3. Study Selection

The search results were exported to EndNote (https://www.myendnoteweb.com, accessed on 10 March 2025) for citation management, and duplicate entries were removed either automatically or manually. Titles and abstracts were screened for relevance, and full-text reviews were performed as needed to confirm eligibility. Only studies utilizing in vivo animal models treated with alkylating agents were considered. Given the limited number of relevant studies in the current literature, all investigations involving the co-treatment with melatonin and alkylating agents were included. Any disagreements regarding study eligibility were resolved through consensus among the authors.

## 3. Results

### 3.1. Selection and Identification of Relevant Studies

Following both algorithmic and manual screening of the selected databases, a total of 382 records were initially identified. After the removal of 155 duplicate entries, 227 unique records remained for further evaluation. Among these, 80 were excluded based on title and abstract screening due to irrelevance to the research topic, and an additional 77 records were excluded because they were in vitro or clinical studies, which did not meet the inclusion criteria.

Further exclusions were made during the full-text assessment. Studies were excluded if they did not investigate outcomes related to the respiratory system, lacked melatonin treatment, or were otherwise unrelated to the defined scope of the review.

Ultimately, only three studies met all the predefined inclusion criteria and were included in the final analysis. A detailed flowchart illustrating the study selection process is provided in Figure 1.

### 3.2. Characteristics of Studies

Table 1 summarizes the key characteristics of the three studies included in this review. While the route of melatonin administration and the number of animals per group was consistent across studies, notable differences were observed in the study designs and dosing regimens. Melatonin was administered at doses ranging from 2.5 to 200 mg/kg/day, with treatment protocols varying from pre-treatment to co-treatment and post-treatment. The duration of treatment ranged from 3 to 7 days.

All the included studies consistently demonstrate that melatonin mitigates lipid peroxidation, as evidenced by reductions in malondialdehyde (MDA) or thiobarbituric acid-reactive substances (TBARSs), and enhances endogenous antioxidant defenses, including glutathione (GSH), superoxide dismutase (SOD), catalase (CAT), and glutathione peroxidase (GPx), across various alkylating agents and animal models. The protective effects of melatonin were shown to be dose-dependent in both the mechlorethamine (HN2) rat models and the cyclophosphamide (CP) mouse model. In the CP mouse study, the highest melatonin dose (20 mg/kg) restored oxidative stress markers (TBARS, GSH, SOD, and CAT) to levels not significantly different from those of the untreated control group.

In the HN2 rat study by Ucar et al. [16], the higher melatonin dose (40 mg/kg) exerted a more pronounced protective effect on the SOD, GPx, and MDA levels compared to the lower dose (20 mg/kg). Similarly, both studies evaluating the nitrosative stress in HN2-exposed rats reported that melatonin reduced the inducible nitric oxide synthase (iNOS) activity and nitrite/nitrate (NOx) levels in a dose-dependent manner. In both cases, higher melatonin doses (40 mg/kg in Ucar et al. [16] and 100 mg/kg in Macit et al. [25]) brought the NOx levels close to those of the control group, indicating the significant attenuation of nitrosative stress.

Although only one study quantitatively assessed inflammatory cytokines, both studies that conducted histological analyses reported consistent reductions in inflammatory cell infiltration following melatonin treatment. These findings suggest that melatonin exerts a broad anti-inflammatory effect in the context of alkylating-agent-induced toxicity. Notably, Macit et al. [25] found that melatonin was more effective than S-methylisothiourea (SMT), a selective iNOS inhibitor, in reducing cytokine levels, implying that melatonin’s anti-inflammatory activity may extend beyond iNOS inhibition.

The three studies [16,17,25] investigated histopathological changes in lung tissues following exposure to two alkylating agents, CP and HN2. Melatonin treatment consistently preserved the lung architecture and reduced histopathological damage. The improvements included decreased edema, hemorrhage, inflammatory infiltration, epithelial injury, and alveolar septal thickening. These protective effects were dose-dependent, and in some cases, high-dose melatonin restored the lung histology to levels comparable to those in control animals (Table 2).

In all models, melatonin treatment significantly attenuated tissue injury, including inflammation, alveolar damage, and fibrosis, when compared to animals exposed to alkylating agents alone. These findings support melatonin’s potential as a therapeutic agent to mitigate lung injury by alkylating chemotherapy.

A comprehensive understanding of the toxicological profiles of HN2 and CP is crucial for identifying their clinical limitations and the role of antioxidant-based interventions. Table 3 summarizes the spectrum of toxicities associated with these agents, categorized into common toxicities, non-pulmonary toxicities, and pulmonary toxicities. HN2 is primarily associated with bone marrow suppression and gastrointestinal symptoms [26], with the additional non-pulmonary effects including immunosuppression, alopecia, and hearing loss [27]. In contrast, CP is notably linked to hemorrhagic cystitis and cardiotoxicity, as well as non-pulmonary toxicities such as hepatotoxicity and stomatitis [28].

Importantly, both agents induce distinct forms of pulmonary toxicity: HN2 causes respiratory epithelial damage, while CP is associated with pneumonitis and pulmonary fibrosis. These findings underscore the urgent need for targeted protective strategies, such as antioxidant therapies, to counteract the broad range of toxic effects caused by alkylating agents.

### 3.3. Comparative Doses

Numerous in vivo studies support the use of melatonin as a co-treatment to mitigate the adverse effects of alkylating agents [29,30,31,32,33]. These studies suggest that melatonin not only exerts protective effects, but may also enhance the therapeutic outcomes. Specifically, melatonin helps safeguard normal cells from oxidative damage without compromising the cytotoxic efficacy of chemotherapy on malignant cells [34,35]. In addition, melatonin has been shown to reduce mortality and alleviate treatment-related side effects, including cystitis, neurotoxicity, and cardiotoxicity [34].

However, a major challenge lies in translating these promising preclinical findings into clinical practice. One critical discrepancy between animal and human studies is dosing. The melatonin doses used in human trials often differ substantially from the effective doses employed in animal models. To address this issue, we calculated the human equivalent doses (HEDs) based on the effective doses reported in the three selected in vivo studies (Table 4). These calculations were performed using the body surface area (BSA) normalization method, which extrapolates dosage from animal to human models based on body weight and metabolic rate differences [36].

Determining HEDs is a vital step in bridging the gap between preclinical and clinical research. Accurate dose translation helps optimize therapeutic regimens to maximize efficacy while minimizing the risk of adverse effects or subtherapeutic dosing, which could compromise melatonin’s pharmacological potential. Nevertheless, HED estimates have inherent limitations, including interindividual variability in drug metabolism and pharmacodynamics, which must be considered when designing and conducting clinical trials.

Based on our extrapolations, the calculated HEDs derived from the animal studies ranged from 12 mg to 973 mg per day. This highlights a significant discrepancy, as the effective doses used in animal models are often up to 24 times higher than those typically administered in randomized controlled trials (RCTs) involving breast cancer patients [37]. These findings emphasize the need for carefully designed dose-escalation studies to evaluate the safety and efficacy of higher melatonin doses in human populations.

## 4. Discussion

This study demonstrates that melatonin exerts significant protective effects against pulmonary toxicity induced by alkylating agents, as evidenced by improvements in oxidative stress markers, inflammatory mediators, and histopathological outcomes. These findings contribute to the growing body of evidence supporting melatonin’s role as a potent cytoprotective agent in chemically induced organ injury.

However, the number of eligible studies was limited (*n* = 3), restricting our ability to conduct standard review analyses, such as subgroup comparisons, sensitivity analyses, and publication bias assessments. A meta-analysis was also not feasible due to considerable heterogeneity in the study designs, including differences in the animal species, the types of alkylating agents, the melatonin dosages, the administration protocols, and the outcome measures. This variability rendered the data unsuitable for meaningful quantitative synthesis. Consequently, we adopted a qualitative, narrative synthesis. These limitations highlight the current lack of standardized research in this area and underscore the need for more methodologically consistent studies to facilitate future meta-analytical evaluations.

Alkylating agents such as CP and HN2 exert their antineoplastic effects primarily through DNA crosslinking. However, their cytotoxicity is also mediated by the excessive production of RONS; nitric oxide (NO), produced by iNOS; and peroxynitrite, a reactive product of NO and superoxide. These play central roles in tissue injury by damaging cellular macromolecules, including lipids, proteins, and DNA. These processes initiate inflammatory cascades that contribute to off-target toxicity, particularly in pulmonary tissues [26,27,28].

Inflammation is a critical mechanism in alkylating-agent-induced lung toxicity. Exposure to these agents promotes the recruitment of inflammatory cells, such as neutrophils and macrophages, into the respiratory tract [38]. These cells exacerbate tissue damage by releasing pro-inflammatory cytokines, including tumor necrosis factor alpha (TNF-α) and interleukin-1 beta (IL-1β), as well as additional RONS [39]. Increased iNOS expression in lung tissue correlates with heightened inflammatory infiltration and sustained oxidative stress [40].

Melatonin confers protection through multiple mechanisms. It directly scavenges a wide range of reactive species, including hydroxyl radicals, superoxide anions, peroxynitrite, peroxyl radicals, nitric oxide, and singlet oxygen [41]. Due to its amphiphilic nature, melatonin crosses biological membranes and accumulates in various tissues and subcellular compartments [42,43], where it safeguards cellular structures, including membranes, proteins, and DNA, from oxidative damage.

The three studies analyzed [16,17,25] consistently demonstrated dose-dependent reductions in lipid peroxidation (e.g., MDA/TBARs), the restoration of antioxidant enzyme activity (SOD, CAT, and GPx), the normalization of non-enzymatic antioxidants (e.g., GSH), the attenuation of inflammatory cytokines (TNF-α, IL-1β), reduced inflammatory infiltration, and decreased nitrosative stress markers (e.g., iNOS, NOx). These findings support melatonin’s well-documented antioxidant, anti-inflammatory, and nitrosative stress-modulating properties [44,45].

However, the use of different animal models, including Wistar and Sprague Dawley rats and NMRI mice, complicates the interpretation due to fundamental interspecies differences. These include variations in splenic architecture, immune cell distribution, and lung mechanics [46,47]. For instance, mice exhibit a higher alveolar wall area and higher alveolar diameters than rats, potentially making them more susceptible to alveolar–capillary-barrier disruption. Additionally, rats display a broader marginal zone in the spleen, which influences immune responses to alkylating agents and potentially affects melatonin’s protective efficacy.

Despite these differences, all three studies demonstrated that melatonin consistently reduced oxidative and inflammatory markers. Two studies investigated HN2-induced toxicity in rats, while one used a CP-induced model in mice. Melatonin administration significantly lowered the MDA levels, restored antioxidant enzyme activity, and reduced pro-inflammatory mediators such as TNF-α, IL-1β, and NADPH oxidase (NOX). Pre-treatment and post-treatment protocols were both effective, though the dosing and schedules varied across studies (2.5 to 100 mg/kg every 12 or 24 h).

The histopathological findings revealed consistent features of lung injury following alkylating agent exposure, including alveolar damage, edema, hemorrhage, and immune cell infiltration. Melatonin treatment mitigated these changes in a dose-dependent manner. For example, Shokrzadeh et al. [17] reported that 20 mg/kg of melatonin preserved alveolar architecture and reduced inflammatory infiltration, while Ucar et al. [16] and Macit et al. [25] observed decreased edema, hemorrhage, and epithelial injury, ultimately preserving lung morphology and preventing fibrosis.

These effects were mediated by melatonin’s dual action: (i) scavenging RONS to prevent oxidative damage to lipids, proteins, and DNA, and (ii) modulating immune responses to suppress inflammatory infiltration and promote tissue repair. Together, these studies provide strong preclinical evidence supporting melatonin’s efficacy in mitigating alkylating-agent-induced toxicity. However, key translational barriers remain. The differences in dosing regimens and species-specific pharmacokinetics must be addressed. Importantly, melatonin is an endogenous indoleamine with a well-established safety profile, a low cost, and a wide availability [48]. These features make it particularly attractive for both civilian and military applications where the chemical exposure risk is elevated.

Model selection is critical for the study design. For instance, Sprague Dawley rats are optimal for studying fibrosis due to their histopathological similarity to human idiopathic pulmonary fibrosis (IPF), whereas NMRI mice are preferable for acute inflammation studies, and Wistar rats are well-suited for biomarker validation [49,50]. Future research should aim for standardization in histopathological scoring to improve cross-study comparability.

Despite encouraging results, the literature on melatonin’s role in preventing alkylating-agent-induced pulmonary toxicity remains sparse. Further research is urgently needed to validate the current findings and better characterize melatonin’s protective mechanisms. The key directions include the following: (i) Long-term investigations are needed to evaluate melatonin’s potential in preventing or reversing pulmonary fibrosis, focusing on mediators such as transforming growth factor beta (TGF-β) and matrix metalloproteinases (MMPs) [51]. (ii) Advanced molecular analyses (e.g., transcriptomics, proteomics) should explore pathways, including autophagy, endoplasmic reticulum stress, and the epithelial–mesenchymal transition (EMT). (iii) Defining the optimal therapeutic window is essential, particularly when used in combination with chemotherapy to avoid impairing antitumor efficacy [52]. (iv) Studies in tumor-bearing, immunocompetent animals receiving chemotherapy will help evaluate whether melatonin protects lung tissue without reducing the efficacy of alkylating agents.

In parallel, clinical data on melatonin’s use in lung injury and critical care settings further support its potential. High-dose regimens (up to 72 mg/day) have demonstrated safety in the treatment of conditions such as COVID-19 and sepsis [53,54], while also improving sleep and reducing sedation requirements [54]. The DAMSEL2 study identified 20 mg as an optimal dose in sepsis patients [55], providing a useful reference for pulmonary applications. Finally, the broader mechanistic evidence, such as the inhibition of the NLRP3 inflammasome and mitochondrial protection [56,57,58], further supports melatonin’s therapeutic value in respiratory pathology.

## 5. Conclusions

In summary, pulmonary toxicity induced by alkylating agents remains a significant clinical challenge, largely due to the limited availability of effective therapeutic interventions. The underlying pathogenesis is complex and multifactorial, involving direct DNA alkylation, oxidative and nitrosative stress, mitochondrial dysfunction, and the persistent dysregulation of inflammatory and reparative pathways. These mechanisms not only contribute to acute lung injury, but may also predispose to chronic complications, including pulmonary fibrosis.

Melatonin’s unique combination of antioxidant, anti-inflammatory, and cytoprotective properties positions it as a promising candidate for both the prevention and treatment of alkylating-agent-induced lung injury. Preclinical studies have consistently demonstrated that melatonin attenuates oxidative stress, modulates immune responses, and preserves the lung architecture in experimental models. Additionally, its favorable safety profile, affordability, and widespread availability enhance its clinical applicability, particularly in high-risk or resource-limited settings.

By integrating mechanistic insights from basic science with emerging preclinical evidence, this review highlights melatonin’s potential as a therapeutic agent in chemically induced pulmonary toxicity. Nevertheless, significant knowledge gaps remain. Future research should focus on optimizing dosing protocols, identifying the patient populations most likely to benefit, and developing pharmaceutical-grade formulations with a reliable potency and bioavailability.

Taken together, the current evidence supports the continued investigation of melatonin as an adjunctive strategy in oncology and toxicology. Its well-established safety, multifaceted protective mechanisms, and translational potential suggest a valuable role in mitigating lung injury associated with alkylating agent exposure. Ultimately, expanding the therapeutic toolkit with agents such as melatonin may lead to improved clinical outcomes for patients undergoing treatment with alkylating chemotherapeutic drugs.

## Figures and Tables

**Figure 1 antioxidants-14-00712-f001:**
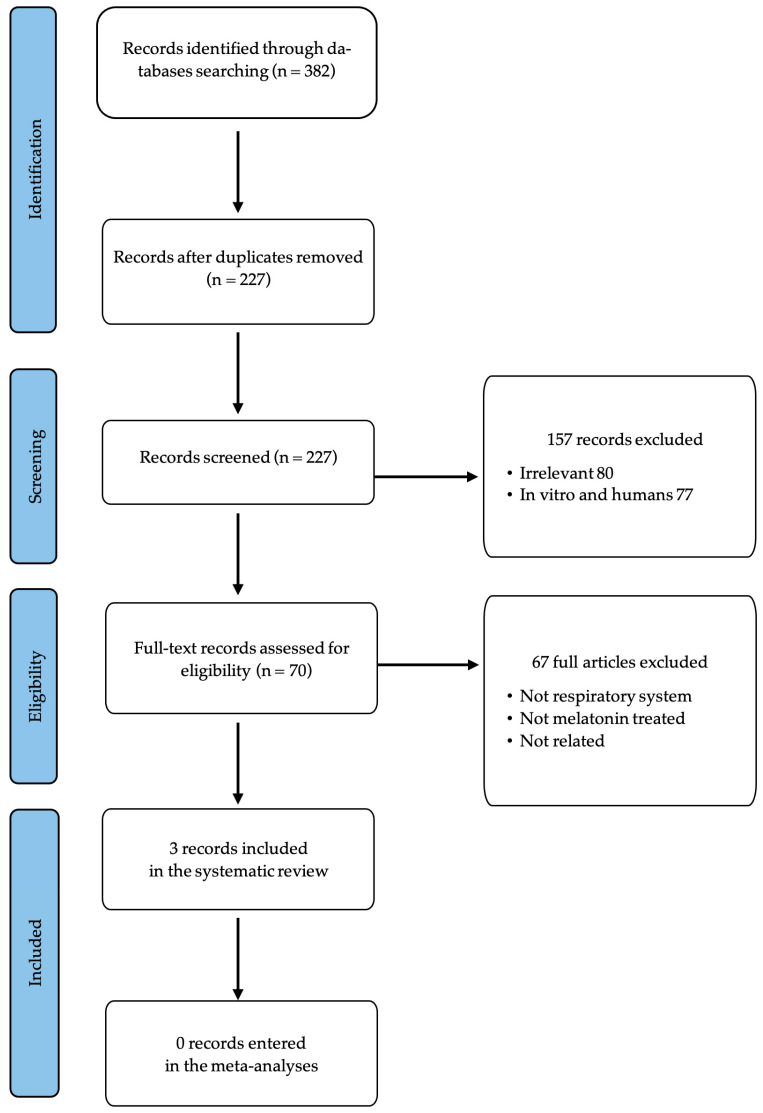
PRISMA flow diagram.

**Table 1 antioxidants-14-00712-t001:** Characteristics of the included in vivo studies.

Reference	Animal Model	Sample Size	Melatonin Dosage (i.p.)	Protocol	AA	Duration	Outcome
[16]	Wistar rats	40	20 or 40 mg/kg every 12 h for 3 days	Co- and post-administration	0.5 mg/kg HN2	5 days	Reduction in lung MDA levels. Promotion of antioxidant enzyme levels. Prevention of GPx decrease.
[25]	Sprague Dawley rats	27	100 mg/kg every 12 h	Co- and post-administration	3.5 mg/kg HN2	3 days	Reduction in proinflammatory cytokines (TNF-α, IL-1β, and NOX).
[17]	NMRI mice	30	2.5, 5, 10, or 20 mg/kg every 24 h	Pre-treatment	200 mg/kg CP	7 days	Reduction in an elevated lipid peroxidation level. Inhibition of GSH depletion. Promotion of antioxidant enzyme activities (SOD, CAT).

AA = alkylating agents; CAT = catalase; CP = cyclophosphamide; GSH = glutathione; GPx = glutathione peroxidase; HN2 = mechlorethamine; IL-1β = interleukin-1β; i.p. = intraperitoneal; MDA = malondialdehyde; NOX = NADPH oxidase; SOD = superoxide dismutase; TNF-α = tumor necrosis factor alpha.

**Table 4 antioxidants-14-00712-t004:** Comparison of the active doses in animal studies and the HED extrapolation for human adults.

Animal Model	Reference	Melatonin Daily Dose	Administration Period	Daily HED for a 60 kg Adult
Mouse	[17]	2.5 mg/kg every 24 h	7 days	12 mg
		5 mg/kg every 24 h	7 days	24 mg
		10 mg/kg every 24 h	7 days	49 mg
		20 mg/kg every 24 h	7 days	97 mg
Rat	[16]	20 mg/kg every 12 h for 3 days	5 days	195 mg
	[16]	40 mg/kg every 12 h for 3 days	5 days	389 mg
	[25]	100 mg/kg every 12 h	3 days	973 mg

HED = human equivalent dose.

**Table 2 antioxidants-14-00712-t002:** Histopathological results of the included studies.

**Reference**	[16]	[25]	[17]
**Protocol**	HN2, 0.5 mg/kg/MLT, 20 or 40 mg/kg every 12 h for 3 days	HN2, 3.5 mg/kg/MLT, 100 mg/kg every 12 h	CP, 200 mg/kg/MLT, 2.5, 5, 10, or 20 mg/kg every 24 h
**Histopathological evaluation after melatonin administration**	Reduction in edema Reduction in alveolar hemorrhage Reduction in inflammatory cell infiltration Improvement of airway pathology	Reduction in alveolar epithelial injury Reduction in inflammation Reduction in interalveolar septal thickening	Reduction in alveolar epithelial injury Reduction in inflammation Reduction in interalveolar septal thickening

HN2 = mechlorethamine; CP = cyclophosphamide; MLT = melatonin.

**Table 3 antioxidants-14-00712-t003:** Summary of common, non-pulmonary, and pulmonary toxicities associated with HN2 [26,27] and CP [28].

Alkylating Agent	Common Toxicities	Non-Pulmonary Toxicities	Pulmonary Toxicities
HN2	Bone marrow suppression Gastrointestinal symptoms Myelosuppression Tissue damage Vesication Skin and mucous membrane damage	Bone marrow suppression Gastrointestinal symptoms Hemorrhagic complications Thrombophlebitis Immunosuppression Alopecia Hearing loss Tinnitus Jaundice Impaired spermatogenesis	Lung complications Respiratory epithelial cell damage Ciliary function inhibition
CP	Bone marrow suppression Hemorrhagic cystitis Immunosuppression Cardiotoxicity Alopecia	Hemorrhagic cystitis Hepatotoxicity Immunosuppression Cardiotoxicity Stomatitis/mucositis	Pneumonitis Pulmonary fibrosis Pulmonary edema Acute interstitial pneumonitis

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
