# Peer review of "In Vivo Evidence of Melatonin’s Protective Role in Alkylating-Agent-Induced Pulmonary Toxicity: A Systematic Review"

_antioxidants, 2025, doi:10.3390/antiox14060712_

Round 1
Reviewer 1 Report
This systematic review aims to critically evaluate the current evidence on melatonin’s protective role against alkylating agent-induced pulmonary toxicity. The relevance of this systematic review is moderate because there are only three experimental in vivo studies on rodents which have been analyzed very correctly by the authors, but these results cannot be easily transferred to humans, so more future clinical data can bring a relevant contribution to this topic.
I consider that the article is well written, and the bibliographic analysis is made following the systematic review methodology outlined in the PRISMA statement. The tables are correct and there is no need for a statistician.
Author Response
Reviewer 1
Major comments
This systematic review aims to critically evaluate the current evidence on melatonin’s protective role against alkylating agent-induced pulmonary toxicity. The relevance of this systematic review is moderate because there are only three experimental in vivo studies on rodents which have been analyzed very correctly by the authors, but these results cannot be easily transferred to humans, so more future clinical data can bring a relevant contribution to this topic.
Response: We sincerely thank the reviewer for their thoughtful and constructive feedback. We fully agree with your observation regarding the limited number of available experimental studies and the inherent challenges in translating preclinical findings to clinical settings. As highlighted in our discussion, this limitation underlines the urgent need for further research, particularly clinical trials, to better understand and confirm melatonin’s potential protective effects in humans.
Reviewer 2 Report
This review article evaluates preclinical evidence from three prior publications regarding the protective effects of melatonin against pulmonary toxicity induced by alkylating agents, specifically cyclophosphamide and mechlorethamine. Recognizing the limitations of current clinical interventions in preventing or reversing alkylating agent-induced lung pathology, this review underscores the urgent need for novel therapeutic strategies. Preclinical rodent studies consistently demonstrate that melatonin exerts dose-dependent protective effects against this toxicity, primarily through the reduction of oxidative stress and the modulation of inflammatory pathways. To significantly advance current knowledge and strongly advocate for melatonin's therapeutic potential in the clinical setting, the author should consider the following:
- The review article highlights the beneficial effects of melatonin in alleviating lung damage induced by alkylating agents, including the reduction of edema, inflammation, and alveolar injury. However, it is crucial to acknowledge that the toxicity of agents like mechlorethamine and cyclophosphamide extends beyond the lungs. For instance, mechlorethamine is notably associated with bone marrow suppression, while cyclophosphamide can lead to hemorrhagic cystitis. To provide a more comprehensive understanding of melatonin's potential in mitigating the broader spectrum of alkylating agent toxicities, the author should include a table summarizing the toxic effects of the discussed alkylating agents (mechlorethamine and cyclophosphamide) beyond pulmonary manifestations. This table should detail common non-pulmonary toxicities such as bone marrow suppression (for mechlorethamine) and hemorrhagic cystitis (for cyclophosphamide).
- The author should expand the literature search to encompass studies investigating melatonin's therapeutic potential against all types of side effects induced by various alkylating agents, as limiting the review to only three studies focused solely on pulmonary toxicity might not accurately reflect the extent of research on melatonin's broader protective effects in the context of alkylating agent chemotherapy. Shifting the focus to melatonin therapy across the entire range of alkylating agent-induced toxicities would likely yield a larger pool of relevant studies and provide a more robust evaluation of melatonin's overall protective capacity. For example, research on melatonin's impact on myelosuppression, gastrointestinal issues, or bladder inflammation caused by alkylating agents should be considered. This broader analysis would strengthen the argument for further clinical investigation into melatonin's role in improving the tolerability and safety of alkylating agent chemotherapy.
- More text and figures elucidating the molecular mechanisms underlying melatonin's protective effects and those of alkylating agents would be beneficial for the reader. The reviewer has noted a previous publication by the same group: "Alkylating Agent-Induced Toxicity and Melatonin-Based Therapies" (Front Pharmacol 2022). The author could provide more advanced information based on this paper.
The reviewer have identified several instances of potential grammatical or typographical errors in the provided text. Here is a examples: Current clinical interventions often fails to fully prevent or reverse these pathological changes
Author Response
Reviewer 2
Major comments
This review article evaluates preclinical evidence from three prior publications regarding the protective effects of melatonin against pulmonary toxicity induced by alkylating agents, specifically cyclophosphamide and mechlorethamine. Recognizing the limitations of current clinical interventions in preventing or reversing alkylating agent-induced lung pathology, this review underscores the urgent need for novel therapeutic strategies. Preclinical rodent studies consistently demonstrate that melatonin exerts dose-dependent protective effects against this toxicity, primarily through the reduction of oxidative stress and the modulation of inflammatory pathways. To significantly advance current knowledge and strongly advocate for melatonin's therapeutic potential in the clinical setting, the author should consider the following:
The review article highlights the beneficial effects of melatonin in alleviating lung damage induced by alkylating agents, including the reduction of edema, inflammation, and alveolar injury. However, it is crucial to acknowledge that the toxicity of agents like mechlorethamine and cyclophosphamide extends beyond the lungs. For instance, mechlorethamine is notably associated with bone marrow suppression, while cyclophosphamide can lead to hemorrhagic cystitis. To provide a more comprehensive understanding of melatonin's potential in mitigating the broader spectrum of alkylating agent toxicities, the author should include a table summarizing the toxic effects of the discussed alkylating agents (mechlorethamine and cyclophosphamide) beyond pulmonary manifestations. This table should detail common non-pulmonary toxicities such as bone marrow suppression (for mechlorethamine) and hemorrhagic cystitis (for cyclophosphamide).
Response: We sincerely thank the reviewer for their valuable and insightful comments. In response, we include a new table and a new paragraph in the revised manuscript that outlines the main systemic toxic effects of these agents, including bone marrow suppression and hemorrhagic cystitis. We believe that this addition will enhance the comprehensiveness of our systematic review and provide a more complete context for evaluating melatonin's potential protective role.
The author should expand the literature search to encompass studies investigating melatonin's therapeutic potential against all types of side effects induced by various alkylating agents, as limiting the review to only three studies focused solely on pulmonary toxicity might not accurately reflect the extent of research on melatonin's broader protective effects in the context of alkylating agent chemotherapy. Shifting the focus to melatonin therapy across the entire range of alkylating agent-induced toxicities would likely yield a larger pool of relevant studies and provide a more robust evaluation of melatonin's overall protective capacity. For example, research on melatonin's impact on myelosuppression, gastrointestinal issues, or bladder inflammation caused by alkylating agents should be considered. This broader analysis would strengthen the argument for further clinical investigation into melatonin's role in improving the tolerability and safety of alkylating agent chemotherapy.
Response: We thank the reviewer for their thoughtful and constructive suggestion. We fully agree that a broader analysis of melatonin's protective effects against the wide range of toxicities induced by alkylating agents, such as myelosuppression, gastrointestinal disturbances, and bladder inflammation, could yield a more extensive pool of studies and provide valuable insights into its overall therapeutic potential. However, the primary motivation for conducting this systematic review was the notable scarcity of studies specifically addressing pulmonary toxicity induced by alkylating agents, despite it being a relevant and potentially serious complication in the oncological setting. As discussed in both the introduction and discussion sections of our review, the majority of existing literature has focused on hematological and gastrointestinal toxicities, whereas pulmonary complications, though acknowledged, have received significantly less research attention. This lack of focused investigation is particularly concerning given that pulmonary toxicity can manifest both acutely and chronically, with clinically significant consequences that may severely affect patients’ quality of life and prognosis. For these reasons, we believe it is crucial to highlight this underexplored area and emphasize the need for further research aimed at understanding the underlying mechanisms, improving early detection, and developing effective therapeutic strategies to prevent or mitigate lung damage associated with alkylating agent treatment.
We appreciate the reviewer’s perspective, and we have added a clarifying statement in the discussion to explicitly acknowledge this rationale and to suggest that future reviews could indeed expand the scope to assess melatonin’s broader protective role across other systems.
More text and figures elucidating the molecular mechanisms underlying melatonin's protective effects and those of alkylating agents would be beneficial for the reader. The reviewer has noted a previous publication by the same group: "Alkylating Agent-Induced Toxicity and Melatonin-Based Therapies" (Front Pharmacol 2022). The author could provide more advanced information based on this paper.
Response: We appreciate the reviewer’s comment. We have already included studies from our research group that analyze the protective action of melatonin against vesicants (sulfur and nitrogen mustards) (PMID: 24035908; PMID: 36829956; PMID: 33920224). These publications provide experimental and mechanistic evidence supporting the efficacy of melatonin in mitigating the toxic effects induced by nitrogen mustards, including its antioxidant, anti-inflammatory, and cytoprotective properties. We believe that the inclusion of these references strengthens the scientific foundation of our manuscript and highlights the relevance of melatonin as a potential therapeutic agent in this context.
Detailed comments
The reviewer have identified several instances of potential grammatical or typographical errors in the provided text. Here is a examples: Current clinical interventions often fails to fully prevent or reverse these pathological changes.
Response: We sincerely thank the reviewer for pointing out the grammatical and typographical errors in the manuscript. We have carefully reviewed the entire text and made the necessary corrections to improve the clarity and accuracy of the language. We appreciate your attention to detail, which has helped us enhance the overall quality of the manuscript.
Reviewer 3 Report
This systematic review is a very appropriate contribution to the field of toxicology in lung models, particularly because it focuses on melatonin as a cytoprotectant. Although it is based on a few studies, it details the use of the PRISMA method correctly. Also, it focuses on describing the antioxidant and anti-inflammatory properties of melatonin in the mitigation of lung damage caused by alkylating agents.
However, I believe that there are areas of opportunity for improvement in their discussion, especially in terms of clearly defining the current knowledge gaps in this field. For example, although the review discusses some studies using both rat and mouse models, it does not sufficiently address how physiological differences between these species might influence the results of the studies discussed. In addition, recommendations for future research work could be more precise; in particular, about model selection (in vitro, in vivo, ex vivo), biomarker prioritization, and translational study design. Including these elements would give greater clarity and practical value to the review.
- (Necessary)
Lines 275–287 / Discussion section: The manuscript discusses the use of rats and mice but does not evaluate the potential impact of interspecies physiological differences on the observed inflammatory or histopathological outcomes. It is recommended that the authors briefly address this issue, especially considering differences in splenic immune response and lung physiology between Wistar/Sprague-Dawley rats and NMRI mice. - (Desirable)
Lines 356–371 / Future directions: The suggestions for future research are valuable but could benefit from a more explicit recommendation regarding which model organism would be more appropriate for translational studies, or under what conditions each might be favored (e.g., for fibrosis vs. acute inflammation endpoints). - (Desirable)
Table 1 / Characteristics of the in vivo studies: It might improve clarity to include a column indicating the species used (rat/mouse), as currently this must be inferred from the strain name. Making this more explicit would help non-specialist readers. - (No statistical concerns):
The data synthesis does not involve statistical testing or meta-analysis. The dose extrapolation uses a standard formula based on body surface area. Therefore, a formal review by a statistician is not required.
Author Response
Reviewer 3
Major comments
This systematic review is a very appropriate contribution to the field of toxicology in lung models, particularly because it focuses on melatonin as a cytoprotectant. Although it is based on a few studies, it details the use of the PRISMA method correctly. Also, it focuses on describing the antioxidant and anti-inflammatory properties of melatonin in the mitigation of lung damage caused by alkylating agents.
However, I believe that there are areas of opportunity for improvement in their discussion, especially in terms of clearly defining the current knowledge gaps in this field. For example, although the review discusses some studies using both rat and mouse models, it does not sufficiently address how physiological differences between these species might influence the results of the studies discussed. In addition, recommendations for future research work could be more precise; in particula, about model selection (in vitro, in vivo, ex vivo), biomarker prioritization, and translational study design. Including these elements would give greater clarity and practical value to the review.
Detailed comments
(Necessary)
Lines 275–287 / Discussion section: The manuscript discusses the use of rats and mice but does not evaluate the potential impact of interspecies physiological differences on the observed inflammatory or histopathological outcomes. It is recommended that the authors briefly address this issue, especially considering differences in splenic immune response and lung physiology between Wistar/Sprague-Dawley rats and NMRI mice.
Response: We sincerely thank the reviewer for this valuable observation. In response to your suggestion, we have incorporated a specific text in the Discussion section addressing the physiological and anatomical differences between species, particularly regarding splenic immune responses and lung structure and mechanics in Wistar/Sprague-Dawley rats versus NMRI mice. We agree that these interspecies differences can significantly influence inflammatory and histopathological outcomes, and their inclusion strengthens the translational relevance of our findings. Thank you again for your constructive contribution, which has helped improve the depth and rigor of our analysis.
(Desirable)
Lines 356–371 / Future directions: The suggestions for future research are valuable but could benefit from a more explicit recommendation regarding which model organism would be more appropriate for translational studies, or under what conditions each might be favored (e.g., for fibrosis vs. acute inflammation endpoints).
Response: We thank the reviewer for this insightful suggestion. Following your indication, we have included a brief paragraph addressing the choice of model organisms for translational studies. We now provide a more explicit recommendation on which species may be better suited for specific endpoints, such as using mice for acute inflammatory responses and rats for studies focused on fibrosis and chronic outcomes. We appreciate your thoughtful feedback, which has helped us refine the translational perspective of our work.
(Desirable)
Table 1 / Characteristics of the in vivo studies: It might improve clarity to include a column indicating the species used (rat/mouse), as currently this must be inferred from the strain name. Making this more explicit would help non-specialist readers.
Response: We thank the reviewer for this helpful suggestion. We would like to clarify that the animal model used in each study is already indicated in the second column of Table 1. This column specifies the species (rat or mouse) alongside the strain name to ensure clarity for all readers, including non-specialists.
Reviewer 4 Report
This manuscript is a systematic review centered on the protective role of melatonin in alkylating agent-induced lung toxicity. The chosen topic has some academic and potential clinical significance: alkylating agent-based drugs or toxins can cause severe lung injury, and exploring whether melatonin, a safe antioxidant, can attenuate such injury could be of value in improving the safety of related treatments. The authors have used a systematic review approach to search and summarize the existing literature from which in vivo experimental evidence has been distilled, which is commendable. The content of the review fills a gap in the field to a certain extent and has some novelty. However, the manuscript also has some shortcomings.
- Only three in vivo studies were ultimately included in this paper, and such a small amount of evidence may affect the robustness of the conclusions of the review. The authors should have acknowledged this limitation more explicitly in the Discussion and emphasized that there is still a paucity of studies on melatonin for the prevention and treatment of lung injury from alkylating agents. Many routine review analyses (e.g., subgroup analyses, publication bias assessments, etc.) may not have been possible due to the small number of included studies, and this should be described and explained in the text. The absence of meta-analysis in the current results is understandable, but authors should clearly indicate the reasons for not performing quantitative synthesis, such as high study heterogeneity, non-combinability of data, or insufficient number of studies. If any studies in related fields were not included (e.g., toxicity studies in other models or studies of similar mechanisms in related drugs), authors are advised to state the reasons for exclusion or consider mentioning them in the Discussion to ensure the comprehensiveness of the synthesis.
-
The authors summarize the key findings of the three included studies and provide tables listing the study characteristics and results. This provides the reader with basic information. However, the results analysis section could still be more systematic and in-depth. It is recommended that the authors summarize the findings of each study thematically in the results narrative, rather than just describing each study individually. Given that a quantitative synthesis is not possible, authors may consider adding qualitative comparisons, such as comparing the magnitude of improvement in similar endpoint indicators across studies, to help readers visualize the degree of consistency in the protective effects of melatonin. The persuasive and scholarly value of the review will be enhanced by analyzing the results more systematically and in depth.
-
In the discussion section, the authors mention that melatonin may act through mechanisms such as antioxidant, anti-inflammatory, and promoting cellular repair, and cite some literature in support. These mechanistic elucidations are consistent with existing knowledge, but could still be more fully discussed in relation to the specific findings of this review. It is suggested that the authors could link the changes in specific indicators observed in the included studies to the underlying mechanisms, for example, how the alterations in oxidative stress markers (e.g., decreased levels of malondialdehyde MDA, increased activity of superoxide dismutase SOD) and inflammatory factors (e.g., decreased levels of TNF-α, IL-1β) in the lung tissues of melatonin-treated animals could support the mechanism of its antioxidant and anti-inflammatory effects. This combination can lead to a more empirical basis for mechanism elucidation.
-
In the discussion, the authors tentatively address the prospect of melatonin's clinical application, mentioning, for example, the conversion of animal experimental doses into human equivalent dose ranges. This discussion is very interesting and reflects the authors' concern for clinical feasibility. To complete this section, it is recommended that the authors further flesh out their assessment of clinical translation. On the one hand, a description of melatonin's clinical safety and experience with established applications could be added. For example, what data on the pre-existing high-dose use of melatonin as a nutraceutical/pharmaceutical in humans informs its application in lung injury prevention and treatment. On the other hand, there is a need to discuss whether such high human equivalent doses are feasible in actual clinical practice: whether they are tolerated by patients, whether there are potential side effects, and how convenient the mode of administration (oral or injectable) is for long-term use. If no clinical trials are currently available for alkylator-induced lung injury, the authors may suggest such studies for the future and point out issues that may need to be addressed during clinical translation (e.g., determination of effective dose ranges, strategies for combining with existing therapies, avoiding compromising antitumor efficacy).
-
The wording and grammar of individual English sentences need to be refined. For example, “duplicates entries” should be changed to “duplicate entries”, and expressions such as “reducing biases and diminishing misunderstandings” are slightly hard. For example, “duplicates entries” should be changed to “duplicate entries”, and expressions such as “reducing biases and diminishing misunderstandings” are a bit stiff, and it is suggested that the wording be optimized to make it more in line with the English academic expression. If “HN2” is used in the text to represent a certain alkylating agent, its chemical name should be stated in the first occurrence.
Author Response
Major comments
This manuscript is a systematic review centered on the protective role of melatonin in alkylating agent-induced lung toxicity. The chosen topic has some academic and potential clinical significance: alkylating agent-based drugs or toxins can cause severe lung injury, and exploring whether melatonin, a safe antioxidant, can attenuate such injury could be of value in improving the safety of related treatments. The authors have used a systematic review approach to search and summarize the existing literature from which in vivo experimental evidence has been distilled, which is commendable. The content of the review fills a gap in the field to a certain extent and has some novelty. However, the manuscript also has some shortcomings.
Detailed comments
Only three in vivo studies were ultimately included in this paper, and such a small amount of evidence may affect the robustness of the conclusions of the review. The authors should have acknowledged this limitation more explicitly in the Discussion and emphasized that there is still a paucity of studies on melatonin for the prevention and treatment of lung injury from alkylating agents. Many routine review analyses (e.g., subgroup analyses, publication bias assessments, etc.) may not have been possible due to the small number of included studies, and this should be described and explained in the text. The absence of meta-analysis in the current results is understandable, but authors should clearly indicate the reasons for not performing quantitative synthesis, such as high study heterogeneity, non-combinability of data, or insufficient number of studies. If any studies in related fields were not included (e.g., toxicity studies in other models or studies of similar mechanisms in related drugs), authors are advised to state the reasons for exclusion or consider mentioning them in the Discussion to ensure the comprehensiveness of the synthesis.
Response: We thank the reviewer for this thoughtful and important observation. We fully acknowledge that the small number of in vivo studies (n = 3) included in our systematic review may limit the overall robustness and generalizability of the conclusions. In response to your suggestion, we have revised the Discussion section to more explicitly emphasize this limitation and to clarify the implications for interpretation and future research. As now stated in the manuscript, the limited number of eligible studies, combined with considerable heterogeneity in terms of species, dosing regimens, administration protocols, and outcome measures, precluded the possibility of performing a meta-analysis or standard review analyses such as subgroup analysis or publication bias assessment. We have now clearly justified the absence of quantitative synthesis by noting the insufficient number of comparable studies and the high degree of methodological variability among them.
Thank you again for your helpful input, which has strengthened the clarity and rigor of our manuscript.
The authors summarize the key findings of the three included studies and provide tables listing the study characteristics and results. This provides the reader with basic information. However, the results analysis section could still be more systematic and in-depth. It is recommended that the authors summarize the findings of each study thematically in the results narrative, rather than just describing each study individually. Given that a quantitative synthesis is not possible, authors may consider adding qualitative comparisons, such as comparing the magnitude of improvement in similar endpoint indicators across studies, to help readers visualize the degree of consistency in the protective effects of melatonin. The persuasive and scholarly value of the review will be enhanced by analyzing the results more systematically and in depth.
Response: Thank you for this observation. In order to help readers visualize the degree of consistency in the protective effects of melatonin We have carried out a qualitative comparison of the three studies, which we have included in the characteristics of studies section.
In the discussion section, the authors mention that melatonin may act through mechanisms such as antioxidant, anti-inflammatory, and promoting cellular repair, and cite some literature in support. These mechanistic elucidations are consistent with existing knowledge, but could still be more fully discussed in relation to the specific findings of this review. It is suggested that the authors could link the changes in specific indicators observed in the included studies to the underlying mechanisms, for example, how the alterations in oxidative stress markers (e.g., decreased levels of malondialdehyde MDA, increased activity of superoxide dismutase SOD) and inflammatory factors (e.g., decreased levels of TNF-α, IL-1β) in the lung tissues of melatonin-treated animals could support the mechanism of its antioxidant and anti-inflammatory effects. This combination can lead to a more empirical basis for mechanism elucidation.
Response: Thank you for your insightful observations and suggestions. We agree that linking the observed changes in specific biomarkers to the underlying mechanisms of melatonin action would strengthen the empirical foundation of our discussion.
In response, we have revised the Discussion section to more explicitly relate the findings of the included studies to the proposed mechanisms of action. Specifically, we now highlight how reductions in oxidative stress markers—such as decreased malondialdehyde (MDA) levels and increased superoxide dismutase (SOD) activity—as well as decreases in inflammatory mediators like TNF-α and IL-1β, provide concrete evidence supporting melatonin’s antioxidant and anti-inflammatory effects in lung tissues. These mechanistic links are now integrated into the narrative to provide a more cohesive and evidence-based interpretation of melatonin's protective role.
In the discussion, the authors tentatively address the prospect of melatonin's clinical application, mentioning, for example, the conversion of animal experimental doses into human equivalent dose ranges. This discussion is very interesting and reflects the authors' concern for clinical feasibility. To complete this section, it is recommended that the authors further flesh out their assessment of clinical translation. On the one hand, a description of melatonin's clinical safety and experience with established applications could be added. For example, what data on the pre-existing high-dose use of melatonin as a nutraceutical/pharmaceutical in humans informs its application in lung injury prevention and treatment. On the other hand, there is a need to discuss whether such high human equivalent doses are feasible in actual clinical practice: whether they are tolerated by patients, whether there are potential side effects, and how convenient the mode of administration (oral or injectable) is for long-term use. If no clinical trials are currently available for alkylator-induced lung injury, the authors may suggest such studies for the future and point out issues that may need to be addressed during clinical translation (e.g., determination of effective dose ranges, strategies for combining with existing therapies, avoiding compromising antitumor efficacy).
Response: Thank you for your thoughtful and constructive feedback. We appreciate your suggestion to further develop the discussion on the clinical translation of melatonin.
In response, we have revised the Discussion section to incorporate a more comprehensive assessment of melatonin’s potential clinical application. Specifically, we now include information on its established safety profile. However, there are currently no clinical trials investigating melatonin's effects on alkylating agent-induced toxicity. Therefore, as suggested by other reviewers, we have emphasized the importance of conducting future studies in this area.
The wording and grammar of individual English sentences need to be refined. For example, “duplicates entries” should be changed to “duplicate entries”, and expressions such as “reducing biases and diminishing misunderstandings” are slightly hard. For example, “duplicates entries” should be changed to “duplicate entries”, and expressions such as “reducing biases and diminishing misunderstandings” are a bit stiff, and it is suggested that the wording be optimized to make it more in line with the English academic expression. If “HN2” is used in the text to represent a certain alkylating agent, its chemical name should be stated in the first occurrence.
Response: Thank you for your careful reading and valuable linguistic suggestions. In response, we have thoroughly revised the manuscript to improve the clarity and fluency of the English language throughout. We appreciate your helpful feedback, which has contributed to improving the overall quality of the manuscript.
Round 2
Reviewer 2 Report
The manuscript now offers a more refined and slightly more useful discussion, primarily through improved clarity, conciseness, and a stronger emphasis on key points and future directions. These improvements make the article suitable for publication. However, minor grammatical revisions should be made before publication.
- Missing The caption for Table 3 states: "Table 3. Toxic effects of HN2 [26,27] and CP [28]".
- Incorrect Source Reference in Discussion section. The sentence introducing the detailed mechanisms of alkylating agents' toxicity, including RONS production and inflammatory cascades, uses "[13]" as a citation. While [13] is a relevant review on melatonin and mustard-induced inflammation, oxidative stress, and DNA damage, it's not the primary source for the general toxic effects of HN2 and CP as categorized in Table 3. The table itself relies on [26,27,28].
- I have found two typos in the text provided. Alkylating agents such as Pand HN2 exert their antineoplastic effects primarily through DNA crosslinking." It should likely be "CP and HN2" (referring to Cyclophosphamide and Mechlorethamine), as these are the agents consistently mentioned throughout the paper. And, ‘’their cytotoxicity is also mediated by the excessive production of RONS an, nitric oxide (NO)’’.
Author Response
Reviewer 2
The manuscript now offers a more refined and slightly more useful discussion, primarily through improved clarity, conciseness, and a stronger emphasis on key points and future directions. These improvements make the article suitable for publication. However, minor grammatical revisions should be made before publication.
- Missing The caption for Table 3 states: "Table 3. Toxic effects of HN2 [26,27] and CP [28]".
Response: We sincerely thank the reviewer for their thoughtful feedback and for acknowledging the improvements made in the manuscript.
Regarding the missing caption for Table 3, we apologize for the oversight. The caption has now been included “Summary of common, non-pulmonary, and pulmonary toxicities associated with HN2 [26,27] and CP [28]”.
2. Incorrect Source Reference in Discussion section. The sentence introducing the detailed mechanisms of alkylating agents' toxicity, including RONS production and inflammatory cascades, uses "[13]" as a citation. While [13] is a relevant review on melatonin and mustard-induced inflammation, oxidative stress, and DNA damage, it's not the primary source for the general toxic effects of HN2 and CP as categorized in Table 3. The table itself relies on [26,27,28].
Response: We thank the reviewer for pointing out the incorrect source reference in the Discussion section. We agree that citation [13] is not the appropriate primary source for the general toxic effects of HN2 and CP as outlined in Table 3.
We have revised the citation in the sentence to accurately reflect the primary sources used in Table 3, now citing references [26,27,28] accordingly.
3. I have found two typos in the text provided. Alkylating agents such as Pand HN2 exert their antineoplastic effects primarily through DNA crosslinking." It should likely be "CP and HN2" (referring to Cyclophosphamide and Mechlorethamine), as these are the agents consistently mentioned throughout the paper. And, ‘’their cytotoxicity is also mediated by the excessive production of RONS an, nitric oxide (NO)’’.
Response: Thank you. It has been corrected.